# Host-Encoded Aminotransferase Import into the Endosymbiotic Bacteria *Nardonella* of Red Palm Weevil

**DOI:** 10.3390/insects15010035

**Published:** 2024-01-05

**Authors:** Ying Huang, Zhen-Feng Feng, Fan Li, You-Ming Hou

**Affiliations:** 1State Key Laboratory of Ecological Pest Control for Fujian and Taiwan Crops, Fujian Agriculture and Forestry University, Fuzhou 350002, China; yinghuang30@163.com (Y.H.); 18605986116@163.com (Z.-F.F.); lf981006@163.com (F.L.); 2Department of Plant Protection, Fujian Province Key Laboratory of Insect Ecology, Fujian Agriculture and Forestry University, Fuzhou 350002, China

**Keywords:** bacteriocyte, endosymbiosis, metabolic integration, *Nardonella*, *Rhynchophorus ferrugineus*, transaminase, tyrosine

## Abstract

**Simple Summary:**

Beetles, among which weevils are the most diverse group, are very successful on land. The thick and hard cuticle of weevils plays an important role in their environmental adaptability. Many weevils carry the bacterial endosymbiont *Nardonella*, which is specialized to produce tyrosine, the key precursor for cuticle formation. The *Nardonella*-encoded tyrosine synthesis pathway is incomplete, lacking the last-step aminotransferase gene. The underlying mechanism of the metabolic integration between *Nardonella* and eukaryotic host cells remains elusive. We identified five aminotransferase genes in the red palm weevil genome, of which only *RfGOT1* and *RfGOT2A* were upregulated in the bacteriocyte. RNA interference of the *RfGOT1* or *RfGOT2A* gene led to a decrease in tyrosine levels in the bacteriome, confirming that these two host-encoded transaminase genes are involved in the final step of the tyrosine synthesis pathway. Notably, trafficking of RfGOT1 and RfGOT2A into the *Nardonella* cytoplasm was observed through immunoelectron microscopy and immunofluorescence experiments. These results indicate that the transaminases encoded by the red palm weevil can be transported to the endosymbiont to complement the tyrosine metabolic pathways of *Nardonella*. Our findings highlight the close relationship and intricate metabolic integration between hosts and endosymbionts.

**Abstract:**

Symbiotic systems are intimately integrated at multiple levels. Host–endosymbiont metabolic complementarity in amino acid biosynthesis is especially important for sap-feeding insects and their symbionts. In weevil–*Nardonella* endosymbiosis, the final step reaction of the endosymbiont tyrosine synthesis pathway is complemented by host-encoded aminotransferases. Based on previous results from other insects, we suspected that these aminotransferases were likely transported into the *Nardonella* cytoplasm to produce tyrosine. Here, we identified five aminotransferase genes in the genome of the red palm weevil. Using quantitative real-time RT-PCR, we confirmed that *RfGOT1* and *RfGOT2A* were specifically expressed in the bacteriome. RNA interference targeting these two aminotransferase genes reduced the tyrosine level in the bacteriome. The immunofluorescence-FISH double labeling localization analysis revealed that RfGOT1 and RfGOT2A were present within the bacteriocyte, where they colocalized with *Nardonella* cells. Immunogold transmission electron microscopy demonstrated the localization of RfGOT1 and RfGOT2A in the cytosol of *Nardonella* and the bacteriocyte. Our data revealed that RfGOT1 and RfGOT2A are transported into the *Nardonella* cytoplasm to collaborate with genes retained in the *Nardonella* genome in order to synthesize tyrosine. The results of our study will enhance the understanding of the integration of host and endosymbiont metabolism in amino acid biosynthesis.

## 1. Introduction

The symbiotic relationships among plants, animals, and microbial communities are universal and profoundly affect the life activities of multicellular organisms [1]. The evolution of eukaryotes is often strongly influenced by bacterial symbionts, and symbiotic relationships can impart new biological characteristics to the host [2]. Bacterial symbionts are commonly associated with insects that feed on plant sap and vertebrate blood [3]. The symbiotic relationship between insects and endosymbionts is interesting and intimate because endosymbionts live in specialized host cells, called bacteriocytes [4], and are individually wrapped by host-derived symbiosomal membranes [5].

Coleoptera is the most species-rich order on land, with weevils (Curculionoidea) being the largest superfamily, comprising approximately 70,000 species [6]. Many weevils are invasive pests that cause significant economic losses in agriculture and forestry. Most weevils feed on nutrient-poor leaves, stems, and roots. Many weevils have specialized organs, called bacteriomes, which carry endosymbionts. These endosymbionts provide nutrients to their hosts and help them to survive in harsh ecological environments [7]. The most widespread endosymbiont lineage is Candidatus *Nardonella* (hereafter referred to as *Nardonella*) [8]. The Gram-negative bacterium *Nardonella* has had a symbiotic relationship with a diverse array of weevils for over 125 million years [9]. Due to strict vertical transmission, the genome of *Nardonella* was significantly reduced to only 0.2 Mb [10]. Genomic sequencing results showed that *Nardonella* has retained genes related to the synthesis pathways of tyrosine and peptidoglycan, while its other metabolic capacities are severely limited [11]. *Nardonella*’s biological role is to provide the host with tyrosine, which is a precursor necessary for the formation and hardening of the insect epidermis [11]. The prosperity of weevils in terrestrial ecosystems is partly due to their highly sclerotized exoskeleton, which preserves their bodies from desiccation, predators, and pathogens [11,12,13,14]. Therefore, *Nardonella* is important for the evolutionary success of weevils.

Eukaryotes generally cannot synthesize large quantities of amino acids to meet their own needs and therefore must obtain amino acids from their diet. However, the amino acids that can be provided by the sap of phloem are very limited [15]. Symbiotic microorganisms typically possess many key catalytic enzymes in the amino acid synthesis pathways, resulting in a pattern in which insects provide nutrients to symbiotic bacteria, and symbiotic bacteria provide amino acids to insects in return [16]. However, there is increasing evidence that, in many cases, the biosynthetic pathways of amino acids do not solely belong to one organism, but rather are composed of key catalytic enzymes encoded by both symbiotic bacteria and their hosts [17,18]. Genome analysis has revealed that *Nardonella*’s tyrosine synthesis pathway is also incomplete, lacking the tyrosine transaminase that catalyzes the final step of the reaction [11]. A series of catalytic enzymes encoded by *Nardonella* can synthesize the tyrosine precursor 4-hydroxy-phenylpyruvate (4-HPP), and the host-encoded transaminase is responsible for converting it to tyrosine, highlighting the close and integrated metabolic cooperation between hosts and endosymbionts. A similar pattern was found in the symbiotic relationship between the invasive ant *Cardiocondyla obscurior* and its endosymbiotic bacterium, *Candidatus* Westeberhardia cardiocondylae, which also provides its host with the tyrosine synthesis precursor 4-HPP [19]. In the symbiotic relationship between the pea aphid (*Acyrthosiphon pisum*) and its obligate symbiont, *Buchnera*, the *Buchnera* genome encodes almost all of the enzymes necessary for multiple essential amino acid synthesis pathways, with the exception of the enzymes required for the final step. The catalytic enzymes for the final step of the reaction are all encoded by the pea aphid. These enzymes include branched-chain aminotransferase for isoleucine, leucine, and valine synthesis, aspartate aminotransferase for phenylalanine synthesis, and phenylalanine 4-monooxygenase for tyrosine synthesis [20,21].

The phenomenon of enzymes encoded by insects and endosymbionts together constituting a complete biosynthesis pathway is universal [22,23], making it necessary for symbiotic partners to exchange multiple substances through symbiosomal membranes for metabolic collaboration. The exchange of metabolic substances usually requires transporters capable of transporting various substances through membranes. The transporter proteins encoded by the host genome are responsible for material transportation on the symbiosomal membrane [24]. For example, the transporter protein ApNEAAT1 is located on the symbiosomal membrane and is capable of transporting amino acids between the pea aphid and *Buchnera* [25]. Interestingly, the fact that a complete biosynthetic pathway is composed of host and endosymbiont genes is similar to the evolutionary outcome of organelles. Host-encoded proteins have been shown to be transported to bacterial endosymbionts through secretory pathways [26,27], similar to the biochemical integration found in mitochondria and plastids. In the bacteriocyte of the mealybug (*Planococcus citri*), a gammaproteobacterium, *Moranella,* lives in the cytoplasm of a betaproteobacterium, *Tremblaya*. The MurF protein, encoded by the *P. citri* genome, is transported to the cytoplasm of *Moranella* to participate in the synthesis of its peptidoglycan [28].

The red palm weevil (RPW), *Rhynchophorus ferrugineus* Olivier, native to South Asia and Melanesia, has spread to Oceania, Asia, Africa, Europe, and the Americas through seedling transport and has become the most destructive invasive pest in palm planting regions worldwide [29]. The genome of *R. ferrugineus* has been sequenced [30]. The genome sequence of *Nardonella* strain NardRF, associated with an Italian population of RPW, was recently reported [31]. The way in which RPWs and *Nardonella* cooperate closely in the synthesis of tyrosine requires further in-depth research. There are theoretically two possible scenarios here: the first is that transaminases encoded by the host can be transported into the *Nardonella* cytoplasm and convert 4-HPP manufactured by *Nardonella* to tyrosine in the bacterial cytoplasm; the second is that 4-HPP is transported from *Nardonella* to the cytoplasm of the bacteriocyte, where it can be catalyzed by RPW transaminases to produce tyrosine. Here, we report the functional characterization of RfGOT1 and RfGOT2A as transaminases in RPW and their trafficking into the *Nardonella* cytoplasm.

## 2. Materials and Methods

### 2.1. Weevil Rearing and Tissue Collection

The wild RPWs were repeatedly collected from a nursery garden of *Phoenix canariensis* in Nan’an City, Fujian Province, China, during July 2022. A laboratory population was established by their offspring. The adults were reared with fresh cut sugarcane stems for feeding and allowed to oviposit under laboratory conditions at 27 ± 1 °C, 75 ± 5% relative humidity, and a 12:12 h (L:D) photoperiod. The sugarcane stems where the eggs were laid were carefully pried open with tweezers. The eggs were collected from the sugarcane using a fine brush and placed in a Petri dish covered with moist filter paper for hatching. The eggs were kept under the following conditions: 27 ± 1 °C, 75% relative humidity, and a photoperiod of 24 h darkness. The newly hatched larvae were then placed individually in plastic boxes with sugarcane stems in a climatic chamber set at 27 ± 1 °C, 75% relative humidity, and a 24 h dark photoperiod.

### 2.2. Characterization of Aminotransferase Genes

The RPW genome (accession: JABAOJ000000000) was subjected to tBLASTn searches using the GOT1A, GOT1B, GOT1C, GOT2A, GOT2B, and TAT protein sequences of *Pachyrhynchus infernalis* as queries, by which the *RfGOT1*, *RfGOT2A*, *RfGOT2B, RfTAT1* and *RfTAT2* gene sequences of RPW were identified. Together with the protein sequences from *Dendroctonus ponderosae*, *Tribolium castaneum*, *Drosophila melanogaster*, *Diorhabda carinulata*, and *Leptinotarsa decemlineata*, multiple alignments were generated using MUSCLE [32]. A phylogenetic tree was constructed with 408 aligned amino acid sites using the neighborjoining and maximumlikelihood methods with the program MEGA7 [33].

### 2.3. RNA Isolation and cDNA Synthesis

Different tissue samples, including the head, gut, fat body, epidermis, and bacteriome, were collected from sixth instar larvae (Appendix A). Before dissection, the larvae were surface-sterilized with 75% ethanol, followed by three rinses in sterilized distilled water. Under a stereomicroscope, tissue dissections were conducted in a clean Petri dish using sterile phosphate-buffered saline (PBS) as well as sterilized scissors and forceps. The tissues from ten larvae were dissected and pooled in 1 mL of sterile PBS as a biological replicate. Samples were frozen in liquid nitrogen and stored at −80 °C until RNA isolation. Total RNA was extracted using TRIzol reagent (Life Technologies, Carlsbad, CA, USA), following the manufacturer’s guidelines. The quality and concentration of the total RNA were determined with 1% agarose gel electrophoresis and a NanoDrop^TM^ 2000 spectrophotometer (Thermo Fisher Scientific Inc., Waltham, MA, USA), respectively. TransScript**^®^** All-in-One First-Strand cDNA Synthesis Super Mix (TransGen, Beijing, China) was used to prepare cDNA and remove genomic DNA contamination.

### 2.4. Relative Transcript Analysis

The qRT-PCR primers for the RPW aminotransferase genes were designed online using PRIMER3 (https://bioinfo.ut.ee/primer3/, accessed on 9 June 2022) and are listed in Table 1. The primers were designed to amplify fragments of approximately 150–200 bp. The amplified products were separated on agarose gel to verify their sizes. *Rfβ-actin* was employed as the reference gene for normalizing target gene expression. The 20 µL reaction system consisted of 10 µL FastStart Universal SYBR Green Master Mix (Roche, Branchburg, NJ, USA), 8 µL sterilized ddH_2_O, 0.5 µL of each primer (10 µM), and 1 µL cDNA. The thermal cycling conditions were 10 min at 95 °C, followed by 40 cycles of 95 °C for 15 s and 60 °C for 1 min. Each reaction for each sample was performed in three technical replicates and four biological replicates. The melting curve was analyzed to confirm the amplification of a single fragment. The relative expression levels of target genes were calculated using the comparative 2^–ΔΔCt^ method [34]. The results were statistically analyzed using ANOVA with the Tukey method. Statistical analyses were performed using IBM SPSS Statistics for Windows, version 19.0 (IBM Corp., Armonk, NY, USA).

### 2.5. RNA Interference of RfGOT1 and RfGOT2A Genes

For synthesis of double-stranded RNA (dsRNA) targeting RfGOT1, RfGOT2A, and green fluorescent protein (GFP), specific primers (Table 1) attached to the T7 promoter sequence were designed using SnapDragon (https://www.flyrnai.org/cgi-bin/RNAi_find_primers.pl, accessed on 10 June 2022). dsRNA was synthesized using the MEGAscript^®^ RNAi Kit (Thermo Fisher Scientific, Waltham, MA, USA) according to the manufacturer’s instructions. Green fluorescent protein (GFP) served as the control dsRNA for the RNAi experiments. A total of 1 µg dsRNA was injected into the body cavity of sixth instar larvae using a thin glass capillary needle. RNase-free water was injected as a control. The bacteriomes were dissected 48 h after dsRNA delivery. The efficiency of RNA interference was measured by RT-qPCR as described above, and the remaining bacteriomes were subjected to tyrosine analysis as described below. The results were statistically analyzed using ANOVA with the Tukey method.

### 2.6. Tyrosine Detection

A Tyrosine Assay Kit (Abcam, Cat no. #ab185435, Cambridge, UK) was used to measure tyrosine levels following the manufacturer’s protocol. The bacteriomes were dissected from six larvae and pooled together as a replicate, and each treatment comprised four replicates. To investigate the impact of RNAi on tyrosine production, bacteriomes were isolated 48 h after dsRNA delivery. The samples were then transferred into 400 μL ice-cold PBS buffer, homogenized with a pellet pestle, and deproteinized using a 10 kDa spin column (Abcam, Cat no. #ab93349). Standards and samples were prepared in a 96-well plate containing the tyrosine assay reagent and incubated at room temperature for 60 min, protected from light. Then, the absorbance of the samples in the plates was detected at 492 nm using a microplate reader.

### 2.7. Histology

Two custom polyclonal antibodies were generated (ABclonal Biotechnology, Beijing, China) to the peptide sequences from RfGOT1 (156–293 aa) andRfGOT2A (210–328 aa), which were predicted to be exposed surfaces. Antibody responses to the immunizing peptides were confirmed by ELISA by the ABclonal Biotechnology Company. Dissected bacteriomes were fixed in phosphate-buffered 4%paraformaldehyde. Protein immunofluorescence (IF)-RNA fluorescence in situ hybridization (FISH) double labeling was performed as previously described [35] with anti-RfGOT antibodies, Alexa Fluor 488-conjugated goat anti-rabbit secondary antibody, and Cy3-labeled oligonucleotide probes specifically targeting *Nardonella*’s 16S rRNA [9] (Table 1). For immunoelectron microscopy, dissected bacteriomes were fixed, dehydrated, embedded, and processed into ultrathin sections, as previously described [36]. The ultrathin sections were immunogold-labeled with RfGOT1- or RfGOT2A-specific antibodies and goat anti-rabbit IgGs conjugated with 10 nm gold particles. Then, ultrathin sections were observed under a HITACHI H-7650 transmission electron microscope.

## 3. Results

### 3.1. Identification of Tyrosine Transaminase Genes in RPW

As transaminase plays an indispensable role in the tyrosine synthesis pathway, the transaminases that could convert 4-HPP to tyrosine in the RPW bacteriome were analyzed. Five transaminase genes were identified in the genome of RPW, including one glutamate oxaloacetate transaminase 1 (designated *RfGOT1*, accession number KAF7284028), two glutamate oxaloacetate transaminases 2 (designated *RfGOT2A* and *RfGOT2B*, accession numbers KAF7283325 and KAF7287630), and two tyrosine aminotransferases (designated *RfTAT1* and *RfTAT2*, accession numbers KAF7284188-KAF7284189). The five proteins had calculated molecular weights of 48.3 kDa, 47.6 kDa, 45.8 kDa, 46.7 kDa, and 48.5 kDa, respectively. These genes all belonged to aminotransferase gene family-related subgroup I according to the Panther classification system (http://www.pantherdb.org/, accessed on 9 April 2022) and were orthologous to those of *P. infernalis* and other insects based on phylogenetic analyses (Figure 1). RfGOT1 was 86% identical to *P. infernalis*PiGOT1A (accession numberLC260175), 83% identical to PiGOT1B (accession number LC260176), and 51% identical to PiGOT1C (accession number LC260177). RfGOT2A was highly similar (89% identity) to PiGOT2A (accession number LC260178), while RfGOT2B was moderately similar (63% identity) to PiGOT2B (accession number LC260179). In addition, RfTAT1 was 72% identical to PiTAT (accession number LC260180), and RfTAT2 was 55% identical to PiTAT.

### 3.2. Analysis of the Relative Expression of RPWTyrosine Transaminase Genes

Transaminase gene expression levels were analyzed using real-time qPCR in the gut, bacteriocyte, head, fat body, and epidermis tissues. The results revealed that*RfTAT1* was predominantly expressed in the epidermis (Figure 2D), while *RfTAT2* (Figure 2E) and *RfGOT2B* (Figure 2C) were predominantly expressed in the gut. Of the five transaminase genes analyzed, only the transcript levels of *RfGOT1* (Figure 2A) and *RfGOT2A* (Figure 2B) were preferentially expressed in the larval bacteriome. However, they were not bacteriocyte-specific because low expression was also detected in other tissues. Similarly, in *P. infernalis*, among the six transaminase genes, the aminotransferase genes *PiGOT1A* and *PiGOT2A* represented the predominant transcripts in the larval bacteriome and have been found to be involved in *Nardonella*-mediated tyrosine synthesis [11]. We predicted that RfGOT1 and RfGOT2A are functionally important in *Nardonella*’s tyrosine synthesis pathway.

### 3.3. Effect of RfGOT1 and RfGOT2A Silencing by RNAi in RPW

To identify the functions of RfGOT1 and RfGOT2A, the expression levels of these two genes were suppressed by RNAi. The results of the quantitative RT-PCR analysis showed that the expression levels of *RfGOT1* and *RfGOT2A* were significantly reduced 48 h after dsRNA injection compared with that after the dsGFP and RNase-free water treatments (Figure 3). The interference efficiency of *RfGOT1* in the bacteriome and carcass was 82.4% and 95.3%, respectively. The interference efficiency of *RfGOT2A* in the bacteriocyte and carcass was 91.9% and 83.8%, respectively. The bacteriome dissected from larvae that had been injected with dsRNAs targeting *RfGOT1* or *RfGOT2A* revealed significantly reduced tyrosine production compared to dsGFP- and RNase-free water-treated larvae (Figure 3). These results strongly suggested that RfGOT1 and RfGOT2A complement *Nardonella*-mediated tyrosine production in the bacteriome.

### 3.4. Localization of RfGOT1 and RfGOT2A in the Bacteriocyte

The RPW–*Nardonella* system, in which 4-HPP produced by *Nardonella* is converted to tyrosine, is of interest but as of yet unknown. The precursor could be transported to the host cytoplasm to synthesize tyrosine or converted to tyrosine within *Nardonella* cells. Different approaches were used for in situ localization of RfGOT1 and RfGOT2A within the symbiotic cells. Immunofluorescence localization of RfGOT1 and RfGOT2A proteins to the bacteriocyte using anti-RfGOT1 and anti-RfGOT2A antibodies (green) revealed the expression of RfGOT1 and RfGOT2A in bacteriocyte tissue (Figure 4). Nuclei and *Nardonella* were identified by DAPI (blue) and a CY3-labeled probe (red) targeting bacterial 16S rRNA, respectively. RfGOT1 and RfGOT2A were present throughout the bacteriocyte, where they colocalized with *Nardonella* cells. Immunogold transmission electron microscopy demonstrated the localization of RfGOT1 and RfGOT2A in the *Nardonella* cytosol but also in the bacteriocyte cytosol (Figure 5). These results indicated that RfGOT1 and RfGOT2A are localized to the *Nardonella* cytoplasm and may convert 4-HPP to tyrosine in *Nardonella* cells.

## 4. Discussion

Endosymbiosis is the driving force for the evolution of eukaryotic cells [37]. The weevil–*Nardonella* system represents an intimate nutrient symbiotic model. The extremely reduced *Nardonella* genome lacks almost all other metabolic pathway genes, indicating that *Nardonella* has evolved through long-term endosymbiosis toward a minimal cellular entity, similar to mitochondria, plastids, and chromatophores. In symbiotic relationships, the previously independent networks of the host and endosymbiont form homeostatic linkages. *Nardonella* almost entirely depends on metabolites produced by the host. The only metabolic function retained by *Nardonella* is the tyrosine synthesis pathway, revealing its sole biological function as a provider of tyrosine, which also means that bacteriocytes act as factories for tyrosine synthesis [11]. The transaminase in the final step of the *Nardonella* tyrosine synthesis pathway is encoded by the host, highlighting the complex symbiotic metabolic integration. An interesting question in the weevil–*Nardonella* system is where the precursor 4-HPP is converted into tyrosine—the *Nardonella* cytoplasm or bacteriocyte cytoplasm.

We identified five transaminases in the RPW genome, including one glutamate oxaloacetate transaminase 1 (RfGOT1), two glutamate oxaloacetate transaminase 2 (RfGOT2A and RfGOT2B), and two tyrosine aminotransferases (RfTAT1 and RfTAT2). According to the phylogenetic analysis, the RfGOTs and RfTATs are highly homologous with the GOTs and TATs of the black hard weevil (*P. infernalis*). *GOT2A* and *GOT1A*upregulation in the bacteriome of *P. infernalis* is necessary for *Nardonella*-mediated tyrosine synthesis and involved in the formation of the adult cuticle [11]. Similarly, *RfGOT1* and *RfGOT2A* were highly expressed in the bacteriocyte of RPW larvae. Compared to the control group injected with dsGFP, the experimental group injected with dsRfGOT1 or dsRfGOT2A exhibited significantly reduced tyrosine production in the bacteriome as well as significantly decreased expression levels of *RfGOT1* and *RfGOT2A*. These results suggest that the RfGOT1A and RfGOT2A encoded by RPW are also needed for *Nardonella*-mediated tyrosine synthesis in the bacteriome. Given the convergence of features of weevil/*Nardonella* coevolution, we hypothesize that analogs of *RfGOT1* and *RfGOT2A* genes also control the last step of *Nardonella*-mediated tyrosine biosynthesis in the bacteriomes of other weevils.

Our IF-FISH results showed that RfGOT1 and RfGOT2A were present throughout the bacteriome, where they colocalized with *Nardonella* cells. The localization of RfGOT1 and RfGOT2A in *Nardonella* cells was observed with immunoelectron microscopy using the immunogold labeling method. We hypothesized that the presence of transaminase in *Nardonella* cells could increase the local concentration of both enzymes and substrate, accelerate the tyrosine synthesis reaction, and help to improve the efficiency of tyrosine production in the weevil–*Nardonella* system. Moreover, localization of RfGOT1 and RfGOT2A in the bacteriocyte cytoplasm was also observed (Figure 5), indicating that these two proteins may be synthesized in the bacteriocyte cytoplasm and targeted to the *Nardonella* cytoplasm. Trafficking the enzymes encoded by the host into the endosymbiont can complement the metabolic pathways of the endosymbiont, similar to the case of plastids or mitochondria. It can be inferred that this protein localization pattern reflects the unknown mechanism used by RPW to transport proteins to *Nardonella* cells. It is theoretically possible that mRNA, rather than proteins, could be transported to the endosymbiont and translated by endosymbiont ribosomes. This transport mechanism may be able to transfer soluble proteins and RNA but cannot transport membrane-associated proteins due to biophysical limitations. It is currently unclear to what extent protein migration from the host to the endosymbiont occurs in different symbioses. A cell wall enzyme, MurF, produced by *P. citri*, is localized to the insect bacteriocyte and the cytoplasm of the endosymbiont, *Moranella* [28]. In *Medicago truncatula*, nodule-specific cysteine-rich (NCR) peptides and Trx s1 are targeted to nitrogen-fixing bacteroids in nodules and enter the bacterial cytosol [26,38]. As peptide transporters, BclA of *Bradyrhizobium* sp. strains [39], BacA of *Sinorhizobium meliloti* [40], as well as SbmA of *Escherichia coli* [41] can promote the uptake of structurally diverse peptides, including NCR peptides. The RlpA4 protein, encoded by a bacterial gene in the aphid genome, has a canonical eukaryotic signal peptide and is located in *Buchnera* cells [27]. However, a proteomic study on the pea aphid–*Buchnera* symbiotic system suggested that proteins are rarely transported to *Buchnera* cells from the bacteriocyte in this association [42]. In the cercozoan amoeba, *Paulinella chromatophora*, host-encoded proteins crucial for the chromophore, an organelle in the early stage of evolutionary development, are targeted and transported to the chromophore. One type of chromophore-targeted protein is short proteins, which enter the chromophore without the need for transit peptides and rely on the secretory pathway [43]. The other type of chromophore-targeted protein is long proteins, which rely on an N-terminal transit peptide of approximately 200 amino acids for transportation [44]. Some proteins containing a single transmembrane helix are also targeted to chromophores and may regulate membrane permeability [45].

## 5. Conclusions

An increasing body of evidence suggests that some proteins encoded in the insect genome can work together with proteins encoded in the symbiotic bacterial genome to synthesize important metabolites [46]. Gene loss in endosymbionts and the compensation of nuclear-encoded genes facilitate host control of critical pathways, which could be vital for host fitness [47]. The gene losses in *Nardonella* affect the key enzymatic steps of tyrosine biosynthesis. Our data show that the transaminases RfGOT1 and RfGOT2A encoded by RPW can be transported to *Nardonella* cells to participate in tyrosine production, but the specific transport mechanism requires further study. Our data also indicate that tyrosine could be produced entirely within *Nardonella* cells and subsequently transported to the cytoplasm of the host.

## Figures and Tables

**Figure 1 insects-15-00035-f001:**
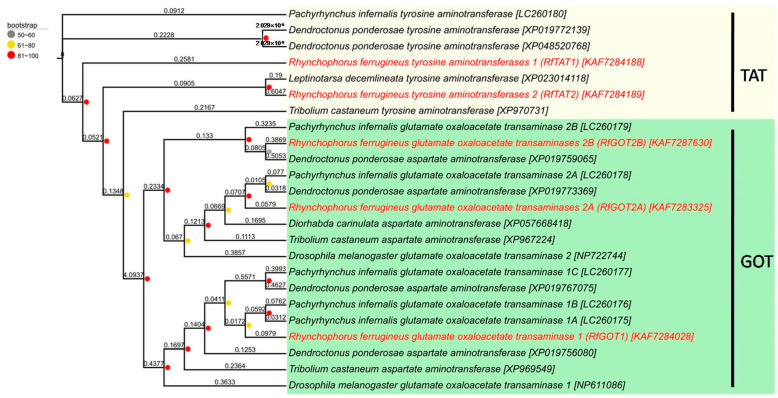
Phylogenetic relationships of transaminase genes identified from *R. ferrugineus* to those from the black hard weevil (*P. infernalis*), the fruit fly (*D. melanogaster*), the flour beetle (*T. castaneum*), the bark beetle (*D. ponderosae*), the Tamarix leaf beetle (*D. carinulata*), and the Colorado potato beetle (*L. decemlineata*). Accession numbers are in brackets. A maximumlikelihood phylogeny inferred from 408 aligned amino acid sites is shown with bootstrap values on the nodes represented as circles. Branch lengths have been displayed. Five transaminase sequences were obtained from *R. ferrugineus*, which are highlighted in red.

**Figure 2 insects-15-00035-f002:**
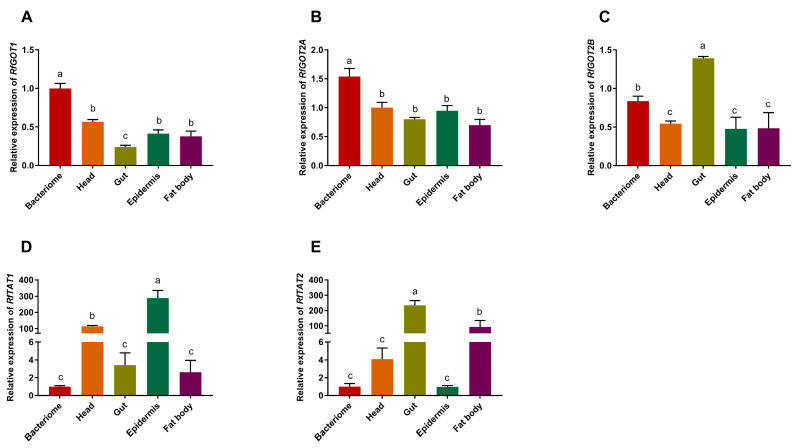
Quantitative gene expression analysis of RPW transaminase genes. Gene expression profiles of RfGOTs and RfTATs were quantified in the bacteriocyte, head, gut, epidermis, and fat body tissues of sixth instar larvae. (**A**) Expression profile of the *RfGOT1* gene; (**B**) Expression profile of the *RfGOT2A* gene; (**C**) Expression profile of the *RfGOT2B* gene; (**D**) Expression profile of the *RfTAT1* gene; (**E**) Expression profile of the *RfTAT2* gene. The transcript level for each transaminase gene was normalized to that of the *β-actin* gene. The error bars represent the standard error, and the different letters indicate significant differences (*p* < 0.05).

**Figure 3 insects-15-00035-f003:**
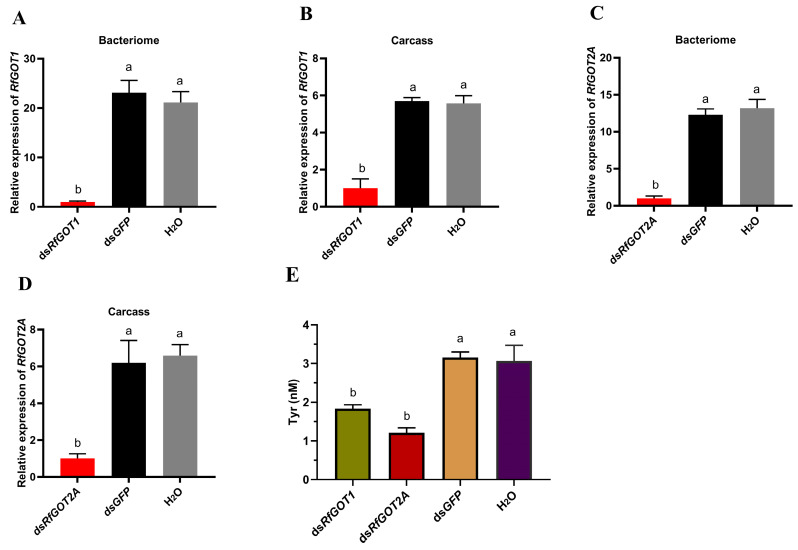
Impact of *RfGOT1* and *RfGOT2A* silencing on RPW larvae. (**A**–**D**) Relative transcript levels of *RfGOT1* and *RfGOT2A* in the bacteriome or carcass (without bacteriome) after RNAi, as detected by RT-qPCR assay; (**E**) Tyrosine levels of larval bacteriome suppressed by RNAi. The error bars represent the standard error, and the different letters indicate significant differences (*p* < 0.05).

**Figure 4 insects-15-00035-f004:**
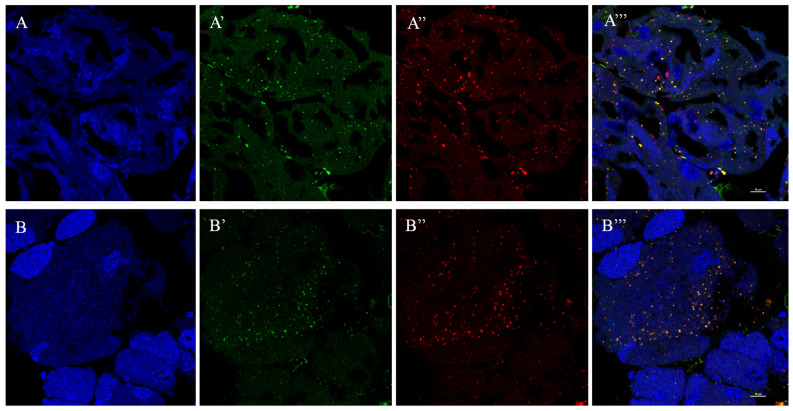
Localization of RfGOT1 and RfGOT2A in the bacteriome as observed with the IF-FISH approach. (**A**–**A**‴) IF-FISH experiments were performed with anti-RfGOT1 antibody; (**B**–**B**‴) IF-FISH experiments were performed with anti-RfGOT2A antibody. (**A**,**B**) DNA was stained with DAPI (blue signal). (**A**′,**B**′) Localization pattern of antibody staining (green signal). (**A**″,**B**″) *Nardonella*’s 16S rRNA was targeted by oligonucleotide probes labeled with CY3 (red signal). (**A**‴,**B**‴) Merged pictures of the antibody images, DAPI-stained DNA, and *Nardonella* probes are shown on the right. (Scale bars: 10 μm).

**Figure 5 insects-15-00035-f005:**
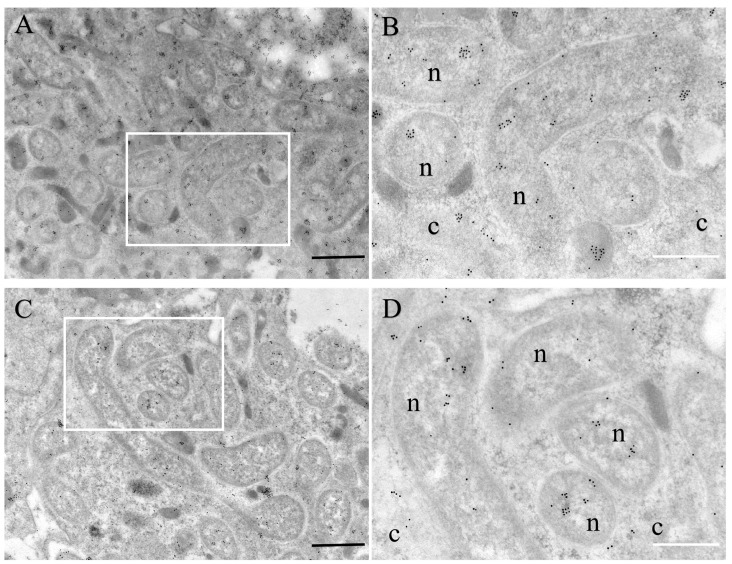
Localization of RfGOT1 and RfGOT2A in the bacteriocyte as observed with immunoelectron microscopy using the immunogold labeling method. (**A**) Localization of RfGOT1 (black dots) in the bacteriocyte; (**C**) Localization of RfGOT2A (black dots) in the bacteriocyte. The scale bar represents 1 μm. (**B**,**D**) are enlarged images of the boxed areas in (**A**,**C**), respectively, and the scale bar represents 500 nm. c, cytoplasm of the bacteriocyte cell; n, *Nardonella* cell.

**Table 1 insects-15-00035-t001:** Primer and probe sequences used in this study.

Primer	Sequence (5′–3′)	Products (bp)
For RT-qPCR		
*RfGOT1*-F	CTTCTGGAGACCTGGATAA	155
*RfGOT1*-R	CACTTCACTTGGGTTGTTC	
*RfGOT2A*-F	GGTTCAACTAAGAGTTGGG	180
*RfGOT2A*-R	TACAGCATGGGCCAGATAG	
*RfGOT2B*-F	CTTTCACGCAGTTGCTCAC	164
*RfGOT2B*-R	ATCGTACCGCCCATACATC	
*RfTAT1*-F	CAAAGCGGTTGCAGAATAT	151
*RfTAT1*-R	GGGTCTGGGTAGCAGAATG	
*RfTAT2*-F	AGGACTGACGGCGTATCTG	181
*RfTAT2*-R	TGGCGTGATTACGATTCTG	
*RfActin*-F	CCAAGGGAGCCAAGCAATT	163
*RfActin*-R	CGCTGATGCCCCTATGTATGT	
For dsRNA synthesis		
*dsRfGOT1*-F	taatacgactcactatagggAGGAGTTGGAGCTTATCGCA	
*dsRfGOT1*-R	taatacgactcactatagggGTTGGATCACATCCGGTAGG	
*dsRfGOT2A*-F	taatacgactcactatagggGTGGGAGCCTATCGAGATGA	
*dsRfGOT2A*-R	taatacgactcactatagggTGCTGGGATCAACACCTGTA	
*dsGFP*-F	taatacgactcactatagggCAGTGCTTCAGCCGCTAC	
*dsGFP*-R	taatacgactcactatagggGTTCACCTGCCGTTCTTGA	
For fluorescence in situ hybridization		
NarYahi933R	AATCTTGCGATCGTACTTCT	
NarYahi1308R	TTCTCGCGAAATTGCTTCT	

The lowercase letters represent the T7 promoter added to the primers for dsRNA synthesis.

## Data Availability

The data presented in this study are available upon request from the corresponding author.

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
