# Peer review of "Host-Encoded Aminotransferase Import into the Endosymbiotic Bacteria Nardonella of Red Palm Weevil"

_insects, 2024, doi:10.3390/insects15010035_

Round 1
Reviewer 1 Report
Comments and Suggestions for Authors
The paper under review considers questions of symbiotic relationships between an insect and a bacterium. Lateral gene transfer and complementation of genetic products derived from both members of a symbiotic system is a widely known phenomenon being under investigation by modern biology all over the world. The reviewed work provides an interesting reading to the journal’s audience. A particular case of an insect and its bacterial symbiont sharing one metabolic pathway is intriguing and deserves in-depth dissection.
The narrative of the introduction is logical and appropriate. Relevant studies are referenced
The methods are explicitly given. I may only recommend the following corrections here:
- a more comprehensive explanation of how the larvae were maintained at the stage when they were introduced into sugarcane stem (L131) – were they still kept in Petri dishes or where?
- a more detailed description on how different tissues were separated, a picture of a dissected insect would be nice to see
- the bacteriomes were dissected from six larvae as a replicate (L180-181), but were they pooled then?
The results are presented straightforwardly. Results of RNAi gene suppression are convincing.
However, the preview version of Figure 1 is not discernible. I have a feeling that some branch support values are below 50. Those might have been hidden for better representation. Neighbor joining doesn’t seem to be necessary here. On the contrary, branch length would have been appropriate to show
Similarly, the resolution of the preview version of Figure 5 is not sufficient to differentiate immunogold labeling specificity inside or outside the bacterial cell
Discussion is sound and logically linked to the results.
Reviewer 2 Report
Comments and Suggestions for Authors
Title of the manuscript: Host-encoded aminotransferase import into the endosymbiotic bacteria Nardonella of red palm weevil.
General comments: This study is significant in determining the role of Nardonella in sclerotizing the exoskeleton of red palm weevil (RPW), Rhynchophorus ferrugineus through the synthesis of tyrosine. This finding can be channeled toward controlling weevils, hence increasing the production of crops attacked by these weevils. Overall, the MS is well-written. However, some of the comments stated below need to be fixed.
Lines 67-69: "The prosperity of weevils in terrestrial ecosystems is partly due to their highly sclerotized exoskeleton which …." I think you should complete this sentence by including how the sclerotization of the exocuticle counts toward their terrestrial prosperity. Is there any reference for this statement? There are other less sclerotized insects that are also very successful on land, so it will be beneficial to clarify that distinction. I think one has to do with their ability to adapt to certain harsh (arid) environmental conditions and not just on land and reducing pathogen invasion through the cuticular surface.
Line 85: "Candidatus Westeberhardia cardiocondylae" only Candidatus should be italicized.
Line 100: "The transporter protein ApNEAAT1 is located on the …"This statement should start with the phrase, 'For example, …'.
Lines 126 to 132: "repeatedly collected" It is not clear if this was done in days, months, or years. It is not clear if RPW was naturally infested in the nursery garden or if eggs or larvae were introduced in the nursery garden.
Line 127: I think the country should be included at the end, "Nan'an City, Fujian Province".
Lines 130-131: "Eggs were removed from the sugarcane and placed in a Petri dish covered with moist filter paper for hatching. The larvae were then introduced into sugarcane stem." How were the eggs removed from the sugarcane and transferred directly to the plant? Were the eggs kept in the incubator at the same climatic conditions as the adults?
Line 138: What sequence length was used for this tree? Was the same length of the genes used for each gene?
Line 143: "Whole insects were also collected." Collected for what? I am presuming you mean that the same thing done for the larvae was repeated in adults. If so, how many replicates? You had 10 replicates for larvae. What about the adults? That statement needs to be clarified.
Line 157: Were the same "cycling conditions" used for all the genes in Table 1?
Line 162: Although mentioned in the result, in this section, I suggest adding the name of the control gene used to normalize the expression.
Line 162: Should the "Turkey method" be the Tukey method?
Line 168: I suggest authors include each gene product's size for RT‒qPCR.
Line 169: The spacing of words in this line is not conforming.
Lines 172-173: "A total of 1 μg dsRNA was injected into the body cavity of sixth instar larvae using a thin glass capillary needle." What was used as the control for explements for this study?
Line 176: The RNA interference study lacks positive control, one of its significant weaknesses.
Line 186: "The absorbance of the plate". It is the absorbance of the solution in the plates (the samples) and not the absorbance of the plate.
Line 223: The quality of the figure is not readable. It is unclear whether the authors used nucleated or amino acids for this tree. What was the length of sequence/amino acids used in Figure 1? Were all genes in the same size?
Lines 232-241: Figure numbers should be put by statements that correspond to them to make it easier to relate figures.
Line 268: Needs to be upper case 'L' in localization.
Lines 285-290: The Figure 4 label does not explain A', A", B' and B".
Comments on the Quality of English LanguageN/A
Reviewer 3 Report
Comments and Suggestions for Authors
Summary: The manuscript by Huang et al aims to test the host-endosymbiont complementarity in Red palm weevil (RPW) and its endosymbiont Nardonella, especially on the tyrosine synthesis pathway. For this, the authors initially identified five aminotransferase genes in the genome of RPW. Later, the expression of two of these genes was confirmed in the bacteriome and recorded the reduction in tyrosine levels in the bacteriome using RNA interference assays of the respective genes. The authors demonstrated the localization of RfGOT1 and RfGOT2A in the bacteriocyte and Nardonella using IF-FISH and Immunogold transmission microscopy. I think the authors have conducted a well thought-out experiment with a good sample size and replication number. The manuscript is well written. I have a few comments for the authors below.
Lines 142-143: Why sixth instar larvae were used and not adult insects?
Line 143: It states, “Whole insects were also collected”. Does it mean adult insects or larvae? Please detail if and how these whole insects were used in the analysis.
Lines 152-153: Since the primers were designed for this study please add a few sentences on the validation of the primers
Lines 180-181: After how many hours of dsRNA injections were the bacteriomes dissected for tyrosine quantification?
Lines 256:-260: Fig 3A and 3C clearly show that dsRNA injections drastically reduced the expression levels of RfGOT1 and RfGOT2A. However, figure 3E shows some levels of production of tyrosine in this bacteriome. Any thoughts about the detection of some levels of Tyrosine in the bacteriome after dsRNA injections? Are there any other known pathways that might contribute to tyrosine production?
Minor comments:
Line 81: spelling: y is missing in 4-hydroxy-
Line 320: spelling “dsGFP”
Lines 322 and 323: Italicization of RfGOT1 and RfGOT2A is inconsistent throughout the manuscript (main text and figure legends). This is true with other gene targets as well. Please correct here and elsewhere in the manuscript wherever it is needed.
